# Energy efficiency to reduce residential electricity and natural gas use under climate change

Janet L. Reyna[1] & Mikhail V. Chester[1]

Climate change could significantly affect consumer demand for energy in buildings, as changing temperatures may alter heating and cooling loads. Warming climates could also lead to the increased adoption and use of cooling technologies in buildings. We assess residential electricity and natural gas demand in Los Angeles, California under multiple climate change projections and investigate the potential for energy efficiency to offset increased demand. We calibrate residential energy use against metered data, accounting for differences in building materials and appliances. Under temperature increases, we find that without policy intervention, residential electricity demand could increase by as much as 41–87% between 2020 and 2060. However, aggressive policies aimed at upgrading heating/cooling systems and appliances could result in electricity use increases as low as 28%, potentially avoiding the installation of new generation capacity. We therefore recommend aggressive energy efficiency, in combination with low-carbon generation sources, to offset projected increases in residential energy demand.

[1] Civil, Environmental, and Sustainable Engineering, Arizona State University, 660 S. College Avenue, Tempe, Arizona 85281, USA. Correspondence and requests for materials should be addressed to J.L.R. (email: janet.reyna@ee.doe.gov).

In the southwestern United States, climate change could lead to particularly large increases in electricity demand via increased need for cooling[1,2]. Increased cooling demands could strain the electricity supply, at the same time that generation and transmission capacity could be affected by decreases in water availability and increases in water and air temperatures[3,4]. Los Angeles County (LAC) has 3.1% of the US national population[5] and consumes 2% of the country's electricity[6]. In addition, a relatively low proportion of homes in the county have air conditioning (40%; ref. 7), so increased adoption of cooling technologies could substantially have an impact on residential electricity demand. The state of California predicts that the population of LAC will increase by 17% to 11.5 million inhabitants by 2060, creating new demands for housing and energy[8]. In the state of California, Executive Order S-3-05 targets greenhouse gas emissions reductions of 80% below 1990 levels by 2050, and previous research suggests that increased electrification of space heating, water heating and transportation coupled with de-carbonization of the electricity supply will be fundamental to meeting this goal[9,10]. Current trends also suggest that consumers are already fuel-switching from fossil fuels to electricity for residential space conditioning[11]. While fuel-switching could help meet greenhouse gas mitigation targets, it will increase demand for electricity, and there is mounting evidence that these traditional power supplies could be constrained by drought and climate change impacts[12,13]. Without proper planning, meeting these demands could come at high economic and environmental costs. Faced with this potentially constrained supply, forecasting demand changes under future climate change is critical for identifying cost-effective paths to meet or mitigate the demand.

Over the past four decades, the state of California has been a leader in pursing aggressive energy efficiency policies, and the state has an ongoing commitment to investing in such programmes. For whole buildings, the California Code of Regulations Title 24 Building Energy Efficiency Standards were first enacted in the 1970s and continue to be updated with increasingly stringent requirements for buildings. More recently, the state has launched several ambitious building energy initiatives, such as the goal of having all new buildings be zero net energy (that is, producing as much energy as they consume) by 2020, and under Assembly Bill 758, the California Energy Commission has been tasked with developing a comprehensive plan to address efficiency in existing buildings, with one of the goals being to double savings from existing building energy efficiency by 2030 (ref. 14). California also had some of the first appliance standards in the United States, established by the newly formed California Energy Commission in the mid-1970s. Title 20 of the California Code of Regulations regulates the efficiency of large appliances, such as refrigerators, water heaters, air conditioners and washing machines, as well as smaller appliances, such as computers, microwaves and light bulbs[15]. All these display an ongoing commitment on the part of the state to invest in progressive energy programmes.

The goal of this study is to forecast residential electricity use in LAC, California, under future climate conditions, and to explore the potential for energy efficiency to offset some of the increased demand. Multiple models exist for energy simulation and forecasting. According to Swan and Ugursal[16], residential sector models fall into one of two larger categories: bottom-up or top-down (Fig. 1). Under this categorization, top-down models are based upon historic, high-level variables such as macroeconomic parameters. The advantages to this modelling approach are availability of data, simplicity of the model and ability to rely upon historic trends in developing a forecast. Top-down models, however, have difficulty forecasting long time horizons, when the base assumptions upon which the model was built have changed (for example, rapidly accelerating population growth, transformational technologies and so on)[17]. Furthermore, top-down models do not subdivide end-use types, which makes it difficult to identify areas for improvement or to understand the physical and behavioural drivers of energy consumption. In contrast, bottom-up models calculate the energy consumption of a subgroup of buildings and then extrapolate to represent the entire building sector. Statistical bottom-up models utilize historic relationships between energy consumption and building end-uses to develop mathematical relationships among the parameters[18], whereas engineering bottom-up models calculate the energy consumption of the end-uses based on the equipment in the buildings without any historic information[19]. The advantage to bottom-up approaches is that end-uses can be directly predicted and targeted for improvement, at the disadvantage of having much greater complexity, data requirements and computation time. Within the engineering bottom-up classification, using building 'archetypes' is a common technique for simulating the electricity use within a city or a city sector[20–29]. Kavgic et al.[19] underscore the lack of available data for identifying technical and behaviour trends for developing quality bottom-up models. To avoid this limitation, many models used for long-term forecasts employ a top-down approach[30]. An emerging group of studies have applied bottom-up archetype and statistical models for climate forecasting[31–36]. An advantage of these studies is that most preserve building end-uses, which is useful for policy recommendations. The majority of these studies simplify the building characteristics to be single zone spaces, estimating thermal loads by balancing heat transfer equations, and do not capture the heterogeneity of the building stock. In addition, the climate change analysis is often a sensitivity analysis to temperature instead of a forecast of future conditions. Dirks et al.[37] develop the forecast with a diverse amount of physical models in EnergyPlus for both residential and commercial buildings across a large portion of the eastern United States with over 26,000 different combinations of technology and building configurations. Climate change is incorporated from a statistically downscaled general circulation model (GCM). Similarly, Wang and Chen[38] use a GCM with two EnergyPlus archetypes representing all of the US residential stock.

For this study, we develop archetype-based bottom-up engineering models to forecast electricity and natural gas (NG) consumption between 2020 and 2060 in the residential sector in LAC. This type of modelling creates a group of building energy simulations that represent the entire building stock. This is the most appropriate model for the project goals as an archetype model is not bound by historic data trends, allows transparency and manipulation of end-use types and allows for the customization of the model to LAC. A drawback to this approach is the high computational resources required to run the building simulation ensemble as well as the detailed data input requirements. We overcome this by simplifying the simulation where possible and utilizing several data sources on building configuration and inhabitant behaviour available for LAC. A bottom-up model is necessary to maintain end-use consumption detail, which we then modify for future years to investigate the impact of energy efficiency improvements across the building stock. An advantage of our modelling over previous efforts is that we utilize a large number of archetypes to capture the heterogeneity of the building stock and also use multiple GCMs and climate scenarios to capture a range of potential climate outcomes. Using this approach, we find that with population growth and projected temperature increases, residential electricity demand could increase by as much as 41–87% between 2020 and 2060. Furthermore, the peak power demand of the most energy-intensive hour of the year could increase by over 220%.

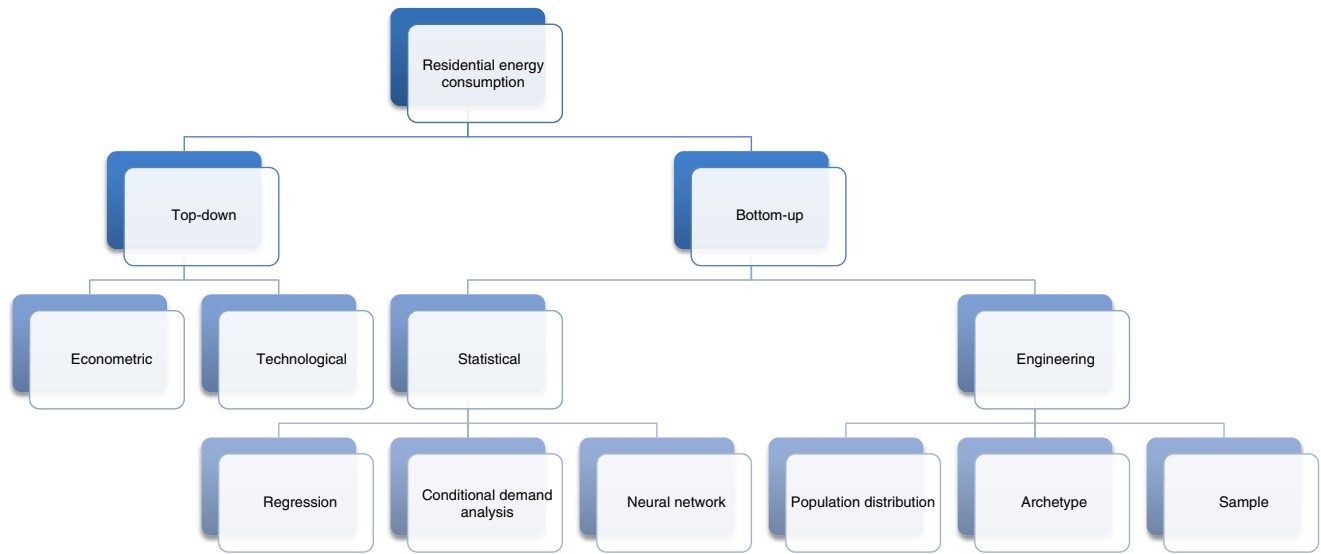

**Figure 1 | Models of residential energy consumption.** Under this schema, the present study is classified as an 'Archetype' model. Adapted from ref. 16.

Aggressive energy efficiency, however, could result in total electricity demand increases of just 28%, and the peak power demand to just 125%. We therefore recommend aggressive energy efficiency, in combination with low-carbon generation sources to offset projected increases in residential energy demand.

## Results

**Increasing energy use under climate change.** Without efficiency intervention, electricity demand in LAC could increase by as much as 87% under Business as Usual (BAU) with heavy electrification (Scenario 2 and Representative Concentration Pathway (RCP) 8.5) or by nearly 47% in BAU without increased electrification (Scenario 1 and RCP 8.5; Fig. 2). Scenario 1 represents an energy future with minimal efficiency increases beyond current policies, and Scenario 2 represents a future that incentivizes heavy electrification of water and space heating without improving efficiency beyond Scenario 1. Under the most optimistic climate, RCP 2.6, electricity demand increases for Scenarios 1 and 2 are 41% and 78%, respectively. We develop these two cases to represent the upper and lower bounds of potential energy savings. In these scenarios, increasing demand is driven by (1) increased adoption of cooling equipment, (2) increased use of cooling equipment as average temperatures increase, (3) population growth and (4) moderate increases in plug loads. These baseline scenarios also include a decrease in NG use, yielding a minimal total energy consumption decrease of 0–1% for Scenario 1 and a decrease of 14–18% for Scenario 2, indicating that without increases in efficiency. These increases are comparable to previous energy forecasts for LAC[10].

**Increasing total demand.** Our results show that, while BAU Scenarios 1 and 2 forecast increases in total energy use under all RCPs, accelerating the adoption of efficient appliances (Scenarios 3 and 4) can offset some or most of the increased demand (Fig. 2). In the two mitigation scenarios under RCP 8.5, energy efficiency can narrow increases in electricity consumption to between 34 and 59%, and total energy use changes between − 26 and − 37%. In Scenario 4 under RCP 2.6 (the most optimistic climate and aggressive policy scenario) electricity use is projected to increase by 34%, and total energy use to decline by 40% (driven by NG reductions). This is over a period with a projected population increase of 17%. This suggests that energy efficiency could be a

viable resource for mitigating increases in energy use. Full results for total demand are in Supplementary Tables 1–8.

**Increasing peak demand.** To ensure reliability, electrical grids must be designed to meet highest demand periods that occur in hot summer months, and our simulations show that the peak demand could more than triple under RCP 8.5 (Fig. 3). This is a larger capacity expansion than the 87% increase in total annual consumption would suggest. Furthermore, previous studies point out that the statistically downscaled weather forecasts tend to underestimate peak temperature increases[39], which means that increases in maximum annual peak demand might be even greater than what is captured in our model. Unlike NG, which is available on-site and on-demand, electricity provision is met from capacity across states, which must be planned in advance and actively managed so that supply can continually meet demand. A scenario with large-scale fuel-switching between electricity and NG could mean an expansion of electricity infrastructure, potentially including the addition of expensive generation stations. A more cost-effective approach might be investing in efficiency; in RCP 8.5, aggressive efficiency reduces the 40-year maximum annual demand increase from 226% to 124%. Efficiency can save money for homeowners[40] and avoid costly upgrades for utility companies[41,42]. For 2060 in Scenario 2 under RCP 8.5, air conditioning is responsible for 85% of electricity consumption at the peak hour; therefore, targeting energy efficiency in cooling equipment (and building thermal shells) could be the most effective approach for reducing peak electricity demand.

**Spatial distribution of changes.** The spatial distribution of electricity demand increases is not uniform throughout LAC and depends upon climate zone, age of building and appliances, and consumption behaviour differences of inhabitants (Fig. 4). The largest percent increases are located in the central inland areas with higher population density (Supplementary Fig. 1) because of the older buildings with aging appliances and poorer insulation as well as the higher temperatures in this area. Inland regions experience the highest increases in temperatures and subsequent energy use under higher RCPs, but under low RCPs these are the areas where net savings could be realized. Location-specific efficiency upgrades, especially with regards to building shell improvements and heating and cooling equipment upgrades,

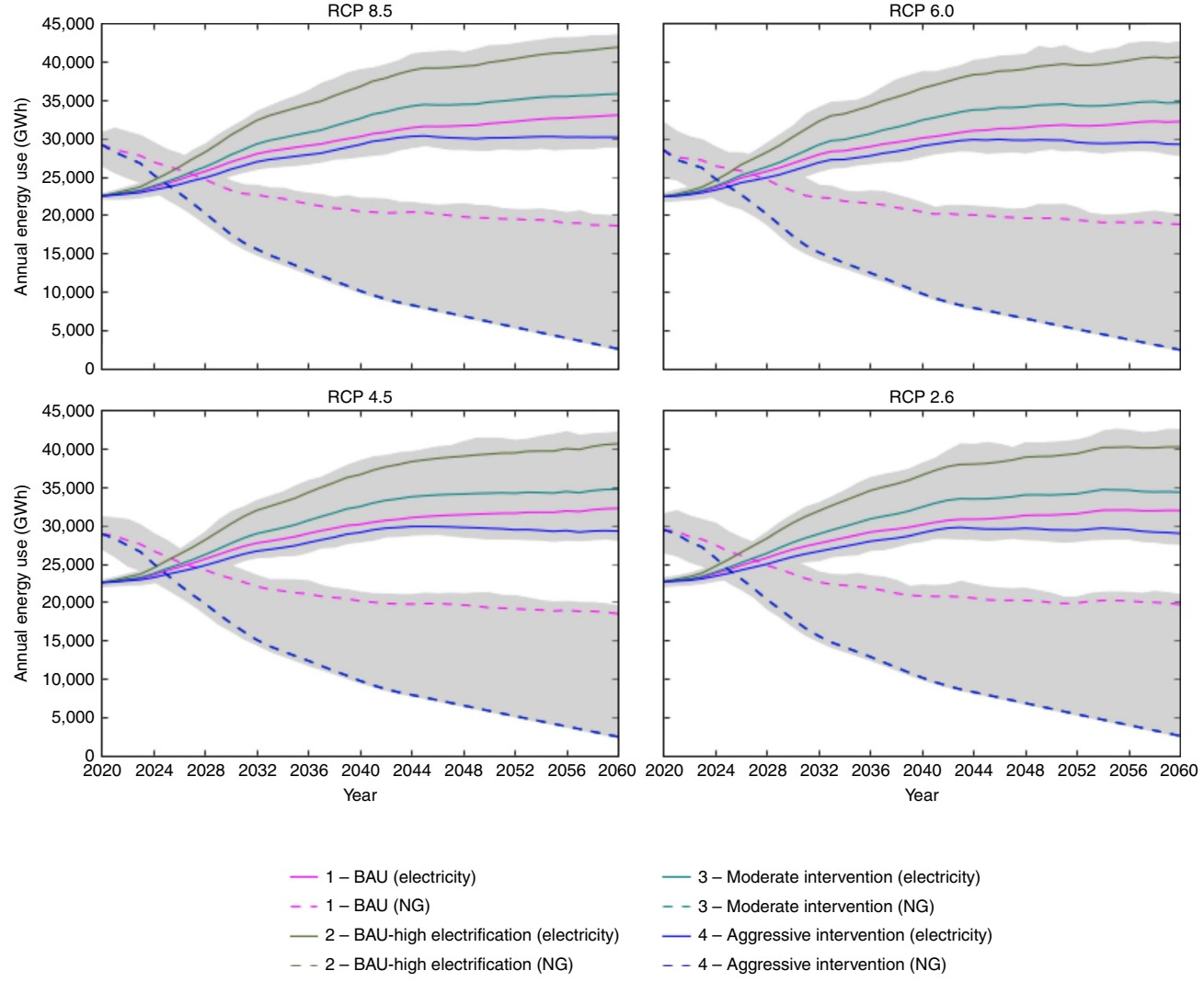

**Figure 2 | Total energy use for all four RCPs and scenarios between 2020 and 2060.** Electricity demand is represented by solid lines and natural gas demand by dashed lines (note: scenarios 2 and 3 are exactly aligned with scenario 4). The grey area shows the variability in our forecast due to the different general circulation models.

could be an effective pathway to maximize energy reduction per dollar invested in efficiency programmes. Spatial results are tabulated in Supplementary Tables 9–13.

## Discussion
Our results show that the majority of projected electricity increases can be offset and net energy use can decline through the aggressive application of energy-efficient technologies. Whereas electricity consumption varies significantly under different RCPs and scenarios, NG consumption is declining in all cases. Within our model, we include all RCPs to represent the potential variability in future climate, but recent studies suggest that current carbon emissions trends most closely follow RCP 8.5 (ref. 43). For RCP 8.5, our results indicate that aggressive energy efficiency (Scenario 4) can save nearly 12 TWh of electricity consumption compared to Scenario 2 in 2060 alone. Space conditioning is a major driver of the increases in electricity consumption in Scenario 2 due to both increased air-conditioning saturation and the electrification of heating technologies. Our results indicate that directly targeting these increases through efficiency could be an effective means of mitigating electricity

increases. Most electricity efficiency savings (~48%) come from the targeting of space heating and cooling: switching the majority of the stock to high-efficiency heat pumps, incentivizing building turnover and improving building thermal shells. In addition, a reduction of 15% can be achieved from efficient heat pump water heaters, 15% from lower consumption of TVs and computers, and 13% from upgraded refrigerators.

Although we focus our research on demand forecasting, increasing prevalence of renewable sources in the energy supply has significant potential to offset projected increases in demand, particularly at peak hours, as well as to alter consumer behaviour. In California, Senate Bill X1-2 sets a goal of having 33% of energy coming from renewable energy sources by 2020, and Senate Bill 350 furthers this to 50% by 2030 (refs 44,45). Although distributed photovoltaics (PV) are excluded from this goal, California will likely continue to provide incentives (such as the California Solar Initiative) for PV in order to reach its greenhouse gas goals. An advantage to PV is that, generally, times of heavy electricity production (that is, when the sun is shining) tend to coincide with periods of high electricity demand driven by heavy air-conditioning use. Because of this, PV has the potential to reduce the load on the electric grid during mid-afternoon peaks

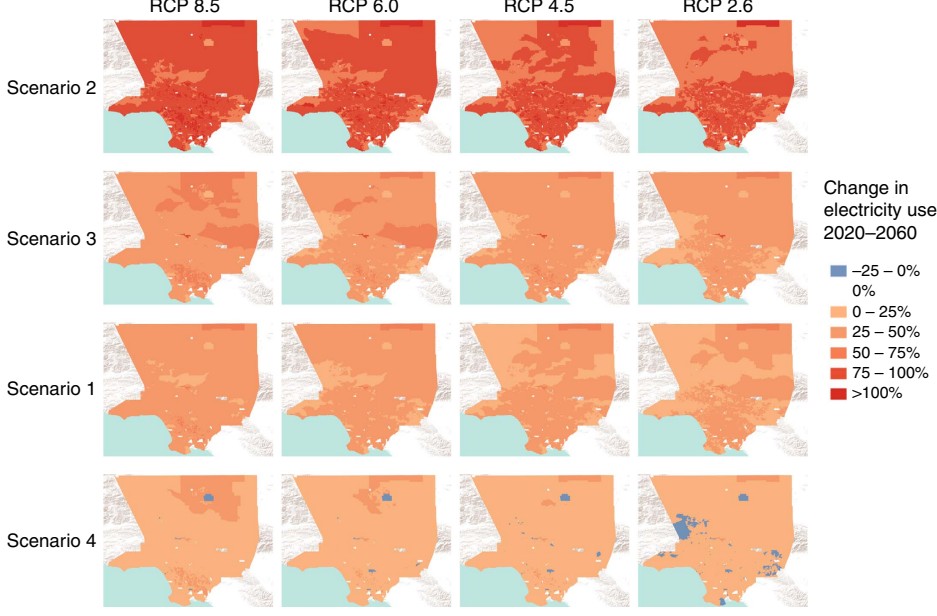

**Figure 3 | Electricity demand 2020–2060.** Here we display the maximum annual electricity demand 2020–2060 and the number of hours per year with an electricity demand above the maximum of 2020. The coloured lines are the average values across all models and the grey shading is the variability due to differences in the GCMs.

**Figure 4 | Change in electricity use across Los Angeles county by census block group 2020–2060.** Each map displays the average per cent change in residential electricity demand for the CBG between 2020 and 2060 under a specific RCP and mitigation scenario. Increases are represented in shades of red and decreases are represented in shades of blue.

and during heat waves, which are also times that PV is most reliable. In fact, California currently has ~9% solar in its generation mix[46], which has effectively offset the traditional afternoon peak and created a new (but lower) peak in the evening after the sun sets[47]. In a situation where a consumer has abundant solar electricity during the day, behaviour might change to shift consumption into the evening hours, exacerbating this new evening peak. Electric vehicles also could be charged in the evening or overnight. These behavioural changes could cause households to consume more energy than in a situation with no solar panels, but some of the additional strain on the electricity grid would be eased regardless, as the afternoon peak could be eliminated, potentially avoiding the installation of new centralized generation facilities. PV could be an effective strategy combined with the energy efficiency measures explored in this paper to prepare for future energy consumption under climate change. Furthermore, other renewable technologies such as solar water heating or on-site electricity storage could be effective pathways to meeting future energy needs. Solar water heating was included in our technology options, but we did not consider large-scale shifting from traditional fuel sources for water heating (that is, electricity and NG) that could potentially lower the demand for water heating significantly.

Aggressive building upgrades and appliance efficiency improvements have the potential to offset projected increases in energy demand. If large-scale fuel-switching between electricity and NG occurs, it will additionally be imperative to reduce consumption as electricity supply could be constrained. Targeting energy efficiency would be most effective if coupled with supply-side strategies such as distributed PV. According to our simulations, in the heating and cooling sector existing technologies could be enough to substantially offset demand increases from fuel-switching; however, timely intervention is necessary to ensure near universal adoption as annual temperatures continue to climb[48]. LAC will have to continually phase-out older technologies and raise the standard for minimum acceptable efficiency, and this will need to be done in stages to avoid large energy and appliance purchase cost increases for consumers. In other appliance categories, aggressive efficiency upgrades will require technology innovation that significantly improves upon commercially existing models. For example, water heaters, which currently consume 28% of energy in LAC, will need to decrease energy consumption by 70% over the next 45 years. Implementation of policies having an impact on these efficiency upgrades will need to happen within the next decade if these aggressive goals are to be met. If such investments are made at the same time as de-carbonizing and improving the resiliency of the energy supply, there could be co-benefits that could significantly lower the greenhouse gas emissions of residential power consumption and reduce costs to consumers. Future research should focus on quantifying the linkages and feedback loops between electricity supply and demand in the presence of renewable energy sources and understanding the cost of implementing different initiatives.

## Methods

**Methods overview.** To quantify the relationship between energy consumption and climate change, we develop a model for forecasting residential energy use between 2020 and 2060 in LAC. Our model is a spatially and temporally resolute bottom-up assessment of residential energy use, which we calibrate against actual consumption data[49]. Using survey data and physical information about the building stock, we create 84 archetype-building simulations in the Building Energy Optimization (BEopt) software to represent all residential buildings in LAC. In BEopt, we utilize EnergyPlus, a state-of-the-art building simulation software developed by the US Department of Energy, as the main simulation engine. We subdivide the archetypes based upon year of construction, classification (that is, single family detached, townhouse and so on) and climate zone. For each archetype we include 21 heating and 13 cooling technologies (Supplementary Table 14). Next,

we scale electricity and NG consumption to the county level while maintaining spatial detail by census block group (CBG). We then calibrate the model using the subset of CBGs that fall within the Los Angeles Department of Water and Power's (LADWP) service area using 1 year of electricity data. An assumption of the model is that patterns of use are correlated to building type, for example, that homeowners in similar vintage homes in the same climate zone will use similar set temps within their homes. We then forecast residential electricity and NG consumption in LAC under climate change and increasing population. We also develop scenarios of varying appliance and building efficiency to investigate the possibility of offsetting projected energy increases.

**Archetype development.** In developing the archetypes, we utilize three major sources of information: the Residential Appliance Saturation Survey (RASS), the LAC Assessor database and the California Assessor's Handbook. RASS is a California-specific appliance survey administered by the California Energy Commission that captures a diverse set of variables on building thermal properties and appliance use. The most recent RASS survey, from 2008, contains ~6,500 survey responses for LAC, and we used these responses to inform appliance distribution within each archetype and some material properties of the buildings (Table 1). The LAC Assessor's office maintains a database of every building standing in LAC[50], primarily for tax purposes, and we utilize their information on building size, classification, location and quality in developing the archetypes. In addition, the California Assessor's Handbook provides 'typical' characteristics of buildings in different quality classes, and it exists as a reference for assessing property value in the state of California. We use the handbook as a complement to the Assessor's database and RASS to add in additional details on thermal properties for each. We give a summary of the data from each source in Table 2.

In LAC, the climate varies greatly between coastal and inland regions; therefore, we differentiate archetypes based upon five climate zones (Supplementary Note 1 and Supplementary Table 15). The California Energy Commission developed these zones specifically for buildings for the purposes of compliance with California Title 24 Building Energy Efficiency Standards (Supplementary Fig. 1).

We develop custom building archetypes based upon our previous work[51], which were then subdivided by climate zone, period of construction and residential building type. There are 140 potential archetype categories, as we consider seven major time periods, five climate zones and four building types ($7 \times 5 \times 4 = 140$), but we aggregate this further to ensure sufficient survey responses in RASS for each archetype. This results in a total of 84 archetypes (Table 3).

We next group all of the residential buildings of LAC in the Los Angeles Assessor database into each of the 84 categories. We use characteristics from the Assessor database as specifications for the archetypes (for example, average building size), and the grouping also allows for the final simulation results to be scaled to the county level. For each of the 84 categories, we compiled a profile of the typical building shape (perimeter to area ratio), predominant material in the framing, average size and quality class from the Assessor database (Table 1). California assessors use the quality class designation to indicate greater quality and

## Table 1 | Variables for archetype definition and sources.

| Variable | Source |
| --- | --- |
| Building size (sqft) | Assessor DB |
| Building age | Assessor DB/RASS |
| Number of stories | RASS |
| Number of bedrooms | RASS |
| Presence of garage | RASS |
| Cooling technology and age | RASS |
| Heating technology and age | RASS |
| Window quality | RASS |
| Framing and foundation | Assessor handbook |
| Exterior finishes | Assessor handbook |
| Interior finishes | Assessor handbook |
| Ceiling fans | RASS |
| Temperature set point | RASS |
| Water heater technology and age | RASS |
| Insulation (walls and attic) | RASS |
| Quality class code | Assessor DB |
| Lighting type | RASS |
| Refrigerator type, size and age | RASS |
| Ranges and ovens | RASS |
| Washer and dryer | RASS |
| TVs and PCs | RASS |
| Pool presence | Assessor DB |
| Pool pump/heater fuel | RASS |

DB, Database; RASS, Residential Appliance Saturation Survey.

**Table 2 | Major drivers of increase and savings compared to 2020 by scenario.**

|  | Scenario 1 | | | | Scenario 2 | | | |
|---|---|---|---|---|---|---|---|---|
|  | RCP 2.6 (%) | RCP 4.5 (%) | RCP 6.0 (%) | RCP 8.5 (%) | RCP 2.6 (%) | RCP 4.5 (%) | RCP 6.0 (%) | RCP 8.5 (%) |
| Temperature (versus RCP 2.6) | 0.0 | 2.0 | 2.2 | 4.7 | 0.0 | 3.1 | 3.4 | 7.1 |
| Population growth | 22.5 | 22.5 | 22.5 | 22.5 | 22.5 | 22.5 | 22.5 | 22.5 |
| HVAC | 19.7 | 19.7 | 19.7 | 19.7 | 48.1 | 48.6 | 48.6 | 49.1 |
| Water heating | − 0.6 | − 0.5 | − 0.5 | − 0.5 | 7.8 | 7.4 | 7.4 | 6.9 |
| Plug loads | 30.1 | 27.3 | 27.1 | 24.4 | 30.1 | 28.6 | 28.4 | 26.8 |
| Other appliances | − 29.5 | − 26.7 | − 26.6 | − 23.9 | − 29.1 | − 27.6 | − 27.5 | − 25.9 |
| **Total change** | **42.3** | **44.3** | **44.5** | **46.9** | **79.4** | **82.5** | **82.8** | **86.5** |
|  | Scenario 3 | | | | Scenario 4 | | | |
| Temperature (versus RCP 2.6) | 0.0 | 2.7 | 3.0 | 6.3 | 0.0 | 2.2 | 2.4 | 5.2 |
| Population growth | 22.5 | 22.5 | 22.5 | 22.5 | 22.5 | 22.5 | 22.5 | 22.5 |
| HVAC | 37.0 | 36.4 | 36.4 | 35.8 | 24.6 | 19.2 | 18.9 | 15.5 |
| Water heating | 4.2 | 3.9 | 3.9 | 3.5 | − 0.1 | − 0.1 | − 0.1 | 0.0 |
| Plug loads | 28.3 | 26.0 | 25.8 | 23.4 | 29.8 | 21.3 | 20.8 | 15.5 |
| Other appliances | − 39.4 | − 36.1 | − 35.8 | − 32.6 | − 48.6 | − 40.4 | − 34.0 | − 25.3 |
| **Total change** | **52.7** | **55.4** | **55.6** | **59.0** | **28.1** | **24.7** | **30.6** | **33.3** |

RCP, representative concentration pathway.

**Table 3 | Core archetype divisions and names**

|  | Apartment or condo (2-4 units) | | | | | Apartment or condo (5 + units) | | | | |
|---|---|---|---|---|---|---|---|---|---|---|
|  | CZ 6 | CZ 8 | CZ 9 | CZ 14 | CZ 16 | CZ 6 | CZ 8 | CZ 9 | CZ 14 | CZ 16 |
| <1940 | 6MFS1 | 8MFS1 | 9MFS1 | 14MFS1-7 | 16MFS1-4 | 6MFL1 | 8MFL1-2 | 9MFL1 | 14MFL1-5 | 16MFL1-7 |
| 1940-1949 | 6MFS2-3 | 8MFS2-3 | 9MFS2 | 14MFS1-7 | 16MFS1-4 | 6MFL2 | 8MFL1-2 | 9MFL2 | 14MFL1-5 | 16MFL1-7 |
| 1950-1959 | 6MFS2-3 | 8MFS2-3 | 9MFS3 | 14MFS1-7 | 16MFS1-4 | 6MFL3 | 8MFL3 | 9MFL3 | 14MFL1-5 | 16MFL1-7 |
| 1960-1969 | 6MFS4 | 8MFS4 | 9MFS4 | 14MFS1-7 | 16MFS1-4 | 6MFL4 | 8MFL4 | 9MFL4 | 14MFL1-5 | 16MFL1-7 |
| 1970-1982 | 6MFS5-7 | 8MFS5 | 9MFS5 | 14MFS1-7 | 16MFS5-7 | 6MFL5 | 8MFL5 | 9MFL5 | 14MFL1-5 | 16MFL1-7 |
| 1983-1997 | 6MFS5-7 | 8MFS6-7 | 9MFS6-7 | 14MFS1-7 | 6MFS5-7 | 6MFL6 | 8MFL6 | 9MFL6 | 14MFL6-7 | 16MFL1-7 |
| 1998-2008 | 6MFS5-7 | 8MFS6-7 | 9MFS6-7 | 14MFS1-7 | 6MFS5-7 | 6MFL7 | 8MFL7 | 9MFL7 | 14MFL6-7 | 16MFL1-7 |
|  | Single family detached | | | | | Townhouse, duplex, or row house | | | | |
|  | CZ 6 | CZ 8 | CZ 9 | CZ 14 | CZ 16 | CZ 6 | CZ 8 | CZ 9 | CZ 14 | CZ 16 |
| <1940 | 6SFD1 | 8SFD1 | 9SFD1 | 14SFD1-3 | 16SFD1-3 | 6SFA1-4 | 8SFA1-2 | 9SFA1 | 14SFA1-7 | 16SFA1-7 |
| 1940-1949 | 6SFD2 | 8SFD2 | 9SFD2 | 14SFD1-3 | 16SFD1-3 | 6SFA1-4 | 8SFA1-2 | 9SFA2 | 14SFA1-7 | 16SFA1-7 |
| 1950-1959 | 6SFD3 | 8SFD3 | 9SFD3 | 14SFD1-3 | 16SFD1-3 | 6SFA1-4 | 8SFA3-4 | 9SFA3-4 | 14SFA1-7 | 16SFA1-7 |
| 1960-1969 | 6SFD4 | 8SFD4 | 9SFD4 | 14SFD4 | 16SFD4 | 6SFA1-4 | 8SFA3-4 | 9SFA3-4 | 14SFA1-7 | 16SFA1-7 |
| 1970-1982 | 6SFD5 | 8SFD5 | 9SFD5 | 14SFD5 | 16SFD5 | 6SFA5 | 8SFA5-7 | 9SFA5 | 14SFA1-7 | 16SFA1-7 |
| 1983-1997 | 6SFD6-7 | 8SFD6-7 | 9SFD6 | 14SFD6 | 16SFD6 | 6SFA6-7 | 8SFA5-7 | 9SFA6 | 14SFA1-7 | 16SFA1-7 |
| 1998-2008 | 6SFD6-7 | 8SFD6-7 | 9SFD7 | 14SFD7 | 16SFD7 | 6SFA6-7 | 8SFA5-7 | 9SFA7 | 14SFA1-7 | 16SFA1-7 |

home value. In some cases, this means improved thermal properties as well. For archetypes that are in the same climate zone that have the same predominant quality class and similar floor areas, we combine them to save on computation time. We maintain all 84 archetypes for appliance assessment, but for the building simulations we use a condensed 51 simulation models (Table 4).

For these 51 categories, we develop models in BEopt using data from the three sources on the thermal properties of the building (Table 1). For HVAC technologies, we use 21 different heating technologies and 13 different cooling technologies within each archetype (Supplementary Table 14). We do this to obtain a more representative 'weighted average' for HVAC end-use energy consumption. For example, in RASS, the survey responses indicated that the main heating technology across most archetypes is a NG furnace of 78% efficiency. However, if we only simulate this one technology for all archetypes, we would fail to capture the true variability in heating technologies (and associated energy use) that exists. Instead, we run all of the technologies in BEopt and weight the energy consumption by archetype category based on the RASS survey responses.

BEopt output is in hourly increments (as is the BEopt core simulation engine, EnergyPlus), and we aggregate this to yearly resolution for the calibration to be consistent with the LADWP data. To get the total consumption for the residential building stock, we normalize HVAC end-use consumption per square foot by archetype, and multiply the square footage of each archetype category within each CBG. The RASS survey reports the frequency of use of HVAC equipment by time

of day, and in many archetype categories, a non-negligible percentage of inhabitants own HVAC equipment, but left it off the majority of the time. We utilize this 'non-use' percentage to adjust the typical energy consumption of the archetypes. In addition, we simulate lighting with the 51 simulation archetypes and normalize per square foot, but we maintain appliances at the per archetype level. In the aggregation, appliance types were maintained so that end-use could be ascertained in the final model and tracked in the forecast.

**Calibration.** We run the simulation ensemble for 2011–2012 and develop custom weather files for BEopt for LADWP climatic conditions to be commensurate with the calibration data set. We calibrate against the median annual residential electricity for the LADWP service area, aggregated by CBG. Researchers at the University of California, Los Angele obtained these data as part of a research project with the California Energy Commission[49]. the University of California, Los Angeles has removed some CBGs that might violate the confidentiality of the account holders. In total, there are 2501 CBGs in the data set that can be used (over 90% of LADWP's service area), and there are 6,422 CBGs total in LAC. The LADWP data are for July 2011 through June 2012. BEopt utilizes an EnergyPlus Weather (EPW) file, which includes a range of climatic variables such as temperature, humidity, solar radiation, snow cover, precipitation and rainfall (Supplementary Table 16). We create a custom EPW for each of the five climate

**Table 4 | Simulation archetype divisions and names**

| | Apartment or condo (2-4 units) | | | | | Apartment or condo (5 + units) | | | | |
|---|---|---|---|---|---|---|---|---|---|---|
| | CZ 6 | CZ 8 | CZ 9 | CZ 14 | CZ 16 | CZ 6 | CZ 8 | CZ 9 | CZ 14 | CZ 16 |
| <1940 | 6MFSD-5 | 8MFSD-4 | 9MFSD-5 | 14MFSD-5 | 16MFSD-4 | 6MFLD-5 | 8MFLD-5 | 9MFLD-5 | 14MFLD-5 | 16MFLD-4 |
| 1940-1949 | 6MFSD-5 | 8MFSD-5 | 9MFSD-5 | 14MFSD-5 | 16MFSD-4 | 6MFLD-5 | 8MFLD-5 | 9MFLD-5 | 14MFLD-5 | 16MFLD-4 |
| 1950-1959 | 6MFSD-5 | 8MFSD-5 | 9MFSD-5 | 14MFSD-5 | 16MFSD-4 | 6MFLD-5 | 8MFLD-5 | 9MFLD-6 | 14MFLD-5 | 16MFLD-4 |
| 1960-1969 | 6MFSD-5 | 8MFSD-5 | 9MFSD-6 | 14MFSD-5 | 16MFSD-4 | 6MFLD-6 | 8MFLD-6 | 9MFLD-6 | 14MFLD-5 | 16MFLD-4 |
| 1970-1982 | 6MFSD-6 | 8MFSD-6 | 9MFSD-6 | 14MFSD-5 | 16MFSD-6 | 6MFLD-6 | 8MFLD-6 | 9MFLD-6 | 14MFLD-5 | 16MFLD-4 |
| 1983-1997 | 6MFSD-6 | 8MFSD-6 | 9MFSD-6 | 14MFSD-5 | 16MFSD-6 | 6MFLD-6 | 8MFLD-6 | 9MFLD-6 | 14MFLD-6 | 16MFLD-4 |
| 1998-2008 | 6MFSD-6 | 8MFSD-6 | 9MFSD-6 | 14MFSD-5 | 16MFSD-6 | 6MFLD-7 | 8MFLD-6 | 9MFLD-7 | 14MFLD-6 | 16MFLD-4 |

| | Single family detached | | | | | Townhouse, duplex, or row house | | | | |
|---|---|---|---|---|---|---|---|---|---|---|
| | CZ 6 | CZ 8 | CZ 9 | CZ 14 | CZ 16 | CZ 6 | CZ 8 | CZ 9 | CZ 14 | CZ 16 |
| <1940 | 6SFDD-5C | 8SFDD-5C | 9SFDD-5C | 14SFDD-5C | 16SFDD-7 | 6SFAD-6C | 8SFAD-6C | 9SFAD-6C | 14SFAD-6M | 16SFAD-7 |
| 1940-1949 | 6SFDD-5C | 8SFDD-6C | 9SFDD-6C | 14SFDD-5C | 16SFDD-7 | 6SFAD-6C | 8SFAD-6C | 9SFAD-6C | 14SFAD-6M | 16SFAD-7 |
| 1950-1959 | 6SFDD-5M | 8SFDD-6M | 9SFDD-6M | 14SFDD-5C | 16SFDD-7 | 6SFAD-6C | 8SFAD-8M | 9SFAD-7M | 14SFAD-6M | 16SFAD-7 |
| 1960-1969 | 6SFDD-7M | 8SFDD-7M | 9SFDD-7M | 14SFDD-7M | 16SFDD-7 | 6SFAD-6C | 8SFAD-8M | 9SFAD-7M | 14SFAD-6M | 16SFAD-7 |
| 1970-1982 | 6SFDD-7M | 8SFDD-7M | 9SFDD-7M | 14SFDD-7M | 16SFDD-7 | 6SFAD-7M | 8SFAD-7M + | 9SFAD-7M | 14SFAD-6M | 16SFAD-7 |
| 1983-1997 | 6SFDD-8M + | 8SFDD-7M + | 9SFDD-7M | 14SFDD-7M | 16SFDD-7 | 6SFAD-6M + | 8SFAD-7M + | 9SFAD-7M | 14SFAD-6M | 16SFAD-7 |
| 1998-2008 | 6SFDD-8M + | 8SFDD-7M + | 9SFDD-8M + | 14SFDD-8M + | 16SFDD-8 | 6SFAD-6M + | 8SFAD-7M + | 9SFAD-8M + | 14SFAD-6M | 16SFAD-7 |

**Table 5 | Building turnover rates.**

| Building vintage | Annual turnover rate (%), Scenarios 1 and 2 | Annual turnover rate (%), Scenarios 3 and 4 |
|---|---|---|
| <1940 | 0.04% | 0.4% |
| 1940–1949 | 0.02% | 0.2% |
| 1950–1959 | 0.01% | 0.1% |
| 1960–1969 | 0.03% | 0.3% |
| 1970–1982 | 0.07% | 0.7% |
| 1983–1997 | 0.3% | 3% |
| 1998 + | N/A | N/A |

N/A, not applicable.

zones for this time period, utilizing climatic inputs from local weather stations[52] and publicly available solar radiation databases[53].

Once we run the ensemble with the appropriate weather data and scale it to county level, we are able to extract the subset of CBGs that exist within the LADWP service area. The goals of the calibration are to (1) have the total modelled electricity consumption be equivalent to the LADWP reported consumption and (2) to have the end-use consumption percentages in the model similar to those reported in RASS. To identify archetypes where thermal properties need to be adjusted, we compare normalized heating and cooling electricity consumption by archetype from the model to the end-use consumption reported by RASS for that archetype. RASS models rather than measures end-use consumption, but this is still useful for identifying which archetypes are above or below the expected value. To prioritize which archetypes to modify, we then weight the deviation from RASS by the average total floor area of all buildings mapped to that archetype. Priority is given to archetypes with high coverage by floor area since they have the largest influence on the model. Once we identify the archetypes for modification, we modify thermal properties of the shell within the uncertainty bounds of the input data sources, for example, changing duct efficiency, changing flooring or increasing insulation. For appliances, rather than adjusting the distribution of types within homes, we use linear scaling factors to adjust consumption towards the expected end-use breakdown. The entire calibration procedure is based upon electricity consumption since that are the data that we have for validation, but NG comprises a significant amount of energy use in LAC residential consumption, mostly in water and space heating. NG data are not available for calibration, but we maintain NG results to compliment the electricity modelling. Final calibrated end-use consumption for the base year is located in Supplementary Table 17.

**Climate change projections.** We develop custom EPW files for each of the five climate zones in LAC in order to forecast changes in building performance under GCM temperature changes. These files are the weather data input for the building simulation software, BEopt. In traditional (that is, non-forecasting) applications, EPW files represent a 'typical' year of meteorological activity for a location, by combining historic weather data (often up to 30 years). This can then be used as a standard for predicting and comparing building performance in a single location, given the assumption that the climate of the location is not changing. In order to use EPWs for forecasting, a unique EPW file must be developed for each forecast and year.

The Intergovernmental on Climate Change's fifth assessment report utilizes four different projections of atmospheric carbon concentrations known as RCPs. Each RCP was developed by an independent modelling team and is designated by their year 2100 radiative forcing level. For example, the most optimistic scenario RCP 2.6 was developed by the IMAGE modelling team at the Netherlands Environmental Assessment Agency has a radiative forcing peak at $3.1\,\mathrm{W\,m^{-2}}$ around the year 2050, but by the year 2100 emissions have been reduced enough to have forcing at $2.6\,\mathrm{W\,m^{-2}}$ by 2100 (ref. 54). RCP 4.5 was developed by the MiniCAM team at Pacific Northwest national Laboratory in the United States, and represents a scenario of stabilizing radiative forcing to $4.5\,\mathrm{W\,m^{-2}}$ well before 2100 (ref. 55). RCP 6.0 models radiative forcing stabilizing at $6.0\,\mathrm{W\,m^{-2}}$ right at 2100 and was created by the AIM modelling team at Japan's National Institute for Environmental Studies[56]. The most pessimistic scenario, RCP 8.5, was developed by the MESSAGE team at the International Institute for Applied Systems Analysis in Austria and includes continually increasing greenhouse gas emissions through 2100 (ref. 57). In this study, we use 10 GCMs for each of the four RCPs to capture a range of future climate scenarios that could have an impact on residential energy consumption.

To maintain spatial differentiation between the climate zones, we utilize statistically downscaled CMIP5 (via bias correction with constructed analogues) projections for temperature. For California, these data are available at $12\,\mathrm{km} \times 12\,\mathrm{km}$ resolution every day for the years 1950–2099 from the US Bureau of Reclamation[58]. To obtain a representative temperature forecast for each model run and climate zone, we take all grid points from that run within a climate zone and average them at every point in time. We then 'morph' the daily trajectories to obtain hourly temperature profiles. Belcher et al.[59] first proposed 'morphing' as a method for using a daily climate forecast to create an hourly profile as is necessary for building the simulation software. We morph the temperature trajectories for each model run, using the modification of Sailor[60] to Belcher's originally proposed method:

$$T_{i,\mathrm{EPW,future}} = \frac{\mathrm{DTR_{GCM}}}{\mathrm{DTR_{EPW}}}\left(T_{i,\mathrm{EPW}} - T_{\mathrm{EPW,min}}\right) + T_{\mathrm{GCM,min}} \qquad (1)$$

where $T_{i,\mathrm{EPW,future}}$ is the temperature at any hour in the future, $\mathrm{DTR_{GCM}}$ and $\mathrm{DTR_{EPW}}$ are the diurnal temperature range (difference in the daily maximum and minimum temperatures) for the model and the base file, respectively, $T_{i,\mathrm{EPW}}$ is the temperature of the base file at that hour, $T_{\mathrm{EPW,min}}$ is the minimum for that day in the base file and $T_{\mathrm{GCM,min}}$ is that daily minimum value in the climate model. For our study, the base weather files are the EPWs developed by the CEC for each of the 16 climate zones in the state of California and available as default files for BEopt. Effectively, this morphing transformation matches the maximum and minimum daily temperatures from the GCM and scales the intermediate hours based on the EPW pattern. For each morphed temperature trajectory, we use a 4-h weighted average to smooth discontinuities between days. In total, we create 1,640 EPW files (10 GCMs × 4 RCPs × 41 years) for each of the five climate zones. These files can then be run with our 51 calibrated archetypes. To be included in our simulation, each climate model must accurately predict the average number of

**Table 6 | Summary of scenario parameters and efficiency changes.**

|  | Scenario 1 | Scenario 2 | Scenario 3 | Scenario 4 |
|---|---|---|---|---|
| Building turnover | Current rate[51] | Current rate[51] | Current rate[51] × 10 | Current rate[51] × 10 |
| Population | 22.5% by 2060 (ref. 8) | 22.5% by 2060 (ref. 8) | 22.5% by 2060 (ref. 8) | 22.5% by 2060 (ref. 8) |
| A/C saturation | CDD adoption curve[1] | CDD adoption curve[1] | CDD adoption curve[1] | CDD Adoption Curve[1] |
| A/C turnover | Purchase trends[72] | Purchase trends[72]– electric only | Heat pump incentives | High-efficiency heat pumps |
| Lighting | − 80% by 2060 | − 80% by 2060 | − 90% by 2060 | − 95% by 2060 |
| TV/computer | − 30% by 2060 | − 30% by 2060 | − 50% by 2060 | − 70% by 2060 |
| Water heating | − 20% by 2060, shift to electric from purchase trends | − 20% by 2060, shift to electric for all new DHW | − 40% by 2060, shift to electric for all new DHW | − 60% by 2060, shift to electric for all new DHW |
| Refrigerators | − 30% by 2060 | − 30% by 2060 | − 50% by 2060 | − 70% by 2060 |
| Plug loads | + 1% per year | + 1% per year | + 0.5% per year | + 0.25% per year |
| Freezer | − 30% by 2060 | − 30% by 2060 | − 50% by 2060 | − 70% by 2060 |
| Microwave | − 50% by 2060 | − 50% by 2060 | − 75% by 2060 | − 80% by 2060 |

A/C, air conditioning; CDD, cooling degree day; DHW, domestic hot water heating.

cooling degree days (CDDs) between 1970 and 2000 within plus or minus 10%. We provide a summary of the included GCMs in Supplementary Table 18.

**Forecast scenarios.** The California Department of Finance forecasts that the population of LAC will grow from 9.8 (today) to 11.5 million in 2060 (ref. 8), and this will necessitate the construction of new dwelling units to accommodate the additional inhabitants. Tracking the population growth with Southern California Association of Government's housing forecast through 2030 (which includes changing household size), we develop bi-decadal housing growth rates for LAC (Supplementary Table 19). We apply housing growth rates based on population to all the scenarios, starting with the building stock from the Assessor's database.

In a previous study, we assessed historic building turnover trends in LAC, and developed a model of building turnover based upon initial year of construction[51]. In addition to housing growth, we include these building turnover rates in the stock model by replacing older vintages with newer vintages of the same classification and climate zone. For Scenarios 1 and 2 we utilize the same rates as in our previous paper, and in Scenarios 3 and 4 we augment the turnover to be 10 times the natural rate to represent incentives for building turnover and building shell upgrades (Table 5). We did not model building shell upgrades individually (for example, improved windows, insulation and so on), but instead we use augmented turnover as a proxy for shell upgrades since newer archetypes have more efficient thermal shells. In applying the population growth and building turnover, we spatially distribute the changes based upon the location of existing dwelling units. For population growth, in reality, new construction might be more likely to occur in less populated CBGs rather than densifying existing areas.

We simulate all archetypes under 10 GCMs and 39 types of heating and cooling equipments, and then we average the results to obtain a mean prediction of total energy with the differentiation in the GCMs representing the variability of the forecast. We also maintain spatial resolution in our simulation ensemble to investigate spatial differences in changing energy demand under climate change.

**Simulation logistics.** Utilizing the custom weather files we developed for each of the GCM runs, we simulate our 51 archetype models using batch processing for EnergyPlus. EnergyPlus is the main simulation engine for BEopt; therefore, once we create the models, we can customize the EnergyPlus input files and directly run them in EnergyPlus. This saves on processing time and allows us to customize the simulation output format. We perform a total of 83,640 simulations: 10 GCMs × 4 RCPs × 41 years × 51 archetypes. With the model output, we post process the data with Python and store it in a SQLite database.

In the archetype calibration, we run the models with 28 different heating and cooling technologies, and for the forecasting we included additional 11 technologies (Supplementary Table 20). To run the full ensemble, this would result in 3,261,960 building simulations, which is computationally limiting. To capture the differences in heating and cooling technologies without running this many simulations, we run the full set of technologies for a single GCM and develop a linear relationship between heating/cooling load of the building and the resulting energy consumption by that technology within that archetype category. In our testing, this method captures the actual amount of energy used by the technology within ± 4%. We utilize these factors in post processing to compute energy consumption for each of the 3,261,960 cases without having to run each of the simulations.

Similar to the calibration phase, we calculate a weighted average of HVAC technologies based on the number of dwelling units and the prevalence of the technology within each archetype category. The main difference is that these factors are temporally dynamic in the forecast since technology adoption and the number of dwelling units change over time. To obtain run totals, we average electricity and NG forecasts for all GCM runs within an RCP. The maximum and minimum energy values across all the models for each year are the uncertainty bounds for that RCP.

**Forecast model.** Within the forecast, we evaluate appliance technology adoption and efficiency gains to test the potential to offset increases in demand. We develop four scenarios to test with each of the climate predictions: (1) business as usual, (2) business as usual + high electrification, (3) moderate efficiency intervention and (4) aggressive efficiency intervention. For each of the scenarios, we include dynamic building turnover rates based upon stock modelling in LAC[51], stock expansion due to population growth, appliances (water heaters, televisions, ovens and so on)[7] and heating/cooling equipment[7]. These variables are dynamically changed within each archetype category for each year between 2020 and 2060 so that there is a distinct forecast for electricity and NG in each hour over this time period. For heating/cooling equipment, we use a weighted average consumption of different technologies within each archetypes as opposed to only using the most prevalent technology (for example, NG furnaces for heating), which captures the variability of technologies used throughout LAC (Supplementary Table 21). Scenario 1 includes existing and proposed policies and normal building and appliance turnover. Scenario 2 includes the same assumptions except that all retired water and space heating equipment, regardless of original fuel type, is replaced with an electric version. This scenario captures aggressive electrification that previous studies claim is necessary to meet the 80% greenhouse gas reduction by 2050 (refs 9,10). Scenario 3 includes heavy electrification (as in Scenario 2), but adds in moderate efficiency gains in appliances beyond existing standards and increased building turnover. Scenario 4 also starts with heavy electrification, but includes efficiency and building gains beyond current technologies. In all cases, we include an increased rate of air-conditioning saturation (that is, the proportion of dwelling units with air conditioners), as previous research finds that saturation correlates strongly with temperatures[1]. Our mitigation strategies are based upon augmented versions of existing policies at the state and federal level and apply to both electric and NG appliances.

As part of our analysis, we estimate the cost of conserved energy. The full background can be found in Supplementary Note 2.

In the next section, we provide an overview of the assumptions underlying each scenario including the current policies and publications that support the development of these assumptions. We summarize efficiency measures in Table 6.

**Heating and cooling equipment.** We turnover existing heating and cooling equipment within each archetype category using a distribution based on the age of the equipment (ex: 80% of a technology by 2020, 90% by 2030 and so on). These turnover time frames are constant for all scenarios. We then create a replacement matrix that gives a distribution of technologies that replaces any retired equipment of each category. For example, a 76% efficient NG boiler might be replaced 10% of the time by an electric furnace, 30% of the time by an 80% efficient NG furnace, 10% of the time by an 85% efficient NG furnace, 10% of the time by a SEER 14 Air-Source Heat Pump, 10% of the time by a SEER 15 air-source heat pump and 10% of the time by a SEER 19 Air-Source Heat Pump. In Scenario 1, we base these replacement rates on purchase trends reported in the US Department of Energy's Building Energy Databook[61]. In Scenario 2, we also use the purchase trends, but remove NG and propane as choices for new appliance replacement and instead distribute new purchases only among electric technologies. In Scenario 3, we restrict all heating replacements to be only heat pumps, and in Scenario 4 all heating and cooling equipments are replaced by only the most efficient heat pumps in the model.

**Air-conditioning adoption.** In addition to the retirement of aging HVAC equipment, we also add in additional cooling equipment based upon previous studies on saturation rates of air conditioning and temperature. Sailor and Pavlova[1] developed an empirical relationship between CDD and per cent saturation of air conditioning based upon data from cities throughout the United States.

$$S_y = S_{2010} + (0.00349)^{(-0.00298xCDDy)}(CDD_y - CDD_{2010}) \quad (2)$$

In this equation, $S_y$ is per cent air-conditioning saturation in a given year, $S_{2010}$ is the initial saturation, CDD is the cooling degree days of the future year and $CDD_{2010}$ is the initial number of CDD. We apply this relationship for every climate zone and every year for all four average RCPs to obtain saturation rates for every year. Since CDD can be somewhat variable from year to year in the forecasts, we forward fill saturation rates so that an air-conditioning saturation rate in a future year cannot be less than in a previous year. For example, if 2034 projects fewer CDD than 2033, we apply the 2033 saturation rate to 2034 since those who purchased air conditioners in 2033 will not discard them the next year. We then apply the saturation rates to archetypes for that climate zone in that year.

**Appliances and plug loads.** With the exception of lighting and plug loads, we assume the number of appliances utilized is linearly proportional to the number of dwelling units. We do not alter the distribution of appliances throughout the scenarios, only the energy consumption, with the exception of the 'plug loads' category that captures use by miscellaneous appliances in homes (for example, cell phones, electronics, blenders and so on). This section discusses the per unit changes in appliances, not to the total amount consumed by each category.

**Lighting.** In 2012, the lighting portion of the Energy Independence and Security Act went into effect regulating the power consumption of incandescent light bulbs in the United States[62]. Beyond increasing the efficiency of incandescent light bulbs, this has driven down the cost of alternative light bulbs such as compact fluorescent lighting and light-emitting diodes (LEDs). The US Department of Energy forecasts that these changes, particularly the adoption of LEDs, will reduce residential electricity consumption from lighting by 53% below 2013 levels by 2030 (ref. 63). Extrapolating market penetration of LEDs and other future lighting technologies we apply lighting electricity consumption reductions of 80% per household by 2060 for Scenarios 1 and 2, 90% for Scenario 3 and 95% for Scenario 4.

**Televisions and computers.** The CEC has proposed regulations on computers and monitors that would take effect in 2017 and 2018. This rule would set performance standards for laptops, desktops and monitors as well as target standby energy consumption[64]. The CEC projects that this rule would reduce consumption of desktop computers by 60% and laptops by 10%. In our forecasts, we assume a computer and television energy reduction of 30% by 2060 for Scenarios 1 and 2, of 50% for Scenario 3 and 70% for Scenario 4.

**Water heating.** We include nine different types of water heaters with four different fuels (electric, gas, propane and solar). In the scenario development, energy consumption could be changed within a archetype category by (1) switching to another type of water heater or (2) changing the efficiency of the nine classifications. For type switching in Scenario 1, we utilize purchase profiles of the domestic hot water market[65] and for Scenarios 2 utilize the same distribution, but restrict purchases to include only electric and solar product types. In Scenario 3 we increase the proportion of heat pumps and solar water heaters to 30% of the total replacement stock by 2060, and in Scenario 4 these technologies comprise 70% of the stock with electric tankless systems making up the remaining 30%. In 2015, Department of Energy efficiency standards went into effect for hot water heaters. For small residential water heaters ($<55$ gallons), these regulations will improve efficiency by $\sim 4\%$. For larger units, efficiency will improve by at least 25% (ref. 66). For all water heater types, we extrapolated the efficiency savings to 20% of the total by 2060 for Scenarios 1 and 2, 40% for Scenario 3 and 60% for Scenario 4.

**Refrigerators and freezers.** In fall 2014, new DOE efficiency standards on refrigerators and freezers went into effect, and these standards are anticipated to lead to savings of $\sim 20–30\%$. For scenarios 1 and 2, we set refrigerators and freezers to be 30% more efficient by 2060, and for Scenarios 3 and 4 we augment these savings to 50% and 70%, respectively.

**Plug loads and miscellaneous electronics.** In our calibrated base model, for each unit we use the equation employed in the US Department of Energy's Building America programme for miscellaneous plug load estimation:

$$E_y = 1,108.1 + 180.2n_b + 0.278a \quad (3)$$

Where $E_y$ is the annual electricity use from plug loads, $n_b$ is the number of bedrooms in the dwelling unit and $a$ is the finished floor area of the unit. During calibration, we find that in LAC, the miscellaneous category needs to be scaled 50% to represent consumption as reported in RASS. Scaling the Building America equation to the county level is:

$$E_{a,LAC} = 1.5(N_u(1,108.1 + 180.2N_b) + 0.278A) \quad (4)$$

Where $E_{a,LAC}$ is the total annual electricity consumption for LAC, $N_u$ is the number of dwelling units, $N_b$ is the total number of bedrooms (from the Assessor) and $A$ is the total finished floor area for all dwelling units.

The change in miscellaneous plug loads over the next 40–50 years is uncertain, but the CEC predicts that they will increase 63.9% for the LADWP service area between 2013 and 2026 (ref. 67). This is growth of nearly 5% per year. As miscellaneous plug loads in the base model are $\sim 30\%$ of total use, extrapolating this trend through 2060 yields an electricity consumption profile dominated by plug loads. For Scenarios 1 and 2, we instead include more conservative forecasts of 1% per year. In Scenarios 3 and 4, we assume efficiency improvements in miscellaneous appliances offsets some of this growth to be 0.5% and 0.25% per year for each scenario, respectively.

**Microwaves.** Standards regulating microwave standby efficiency will go into effect in 2016, and they will reduce standby consumption by 75% (ref. 68). On the basis of this, we include energy savings of 50% for microwaves by 2060 in Scenarios 1 and 2, 75% in Scenario 3 and 80% in Scenario 4.

**Rebound penalty.** Efficiency improvements in homes can also lead to changes in energy consumption behaviour, potentially offsetting some of the savings from efficiency. In real-world application, improvements in energy efficiency in homes can lead to the rebound effect—where expected savings on energy are partially offset by changes in inhabitant behaviour. It is well documented that changes in social or economic conditions lead to changes in consumer behaviour[69]. For example, if a homeowner upgrades to a more efficient air-conditioning unit, they might end up using the air conditioning in the home more frequently, as it is less expensive to operate. To capture some of these potential behavioural changes in our model, we include a 10% elasticity efficiency penalty as the rebound effect. Estimates on the appropriate elasticity of residential energy demand vary widely[70], but a review of recent studies suggests that around 10% demand elasticity is appropriate[71]. The true value of the rebound is somewhat uncertain, but is likely small compared to the other variables in the model such as housing stock growth, climate change and changes in electricity prices (Supplementary Discussion).

**Validation.** We validate our model results by developing a simple linear regression relationship between max daily temperature and residential electricity consumption. We then create a forecast with this regression model to compare against our archetype model.

For the regression, we utilize daily maximum temperatures for downtown Los Angeles (weather station code CQT) from between 2006 and 2010, downloaded from the Iowa Environmental Mesonet[52]. The Federal Energy Regulatory Commission Form 714 provides total hourly electricity demand for LADWP over the same time period. Since our model includes only residential buildings, we then estimate the proportion of LADWP sales that are residential using monthly totals of residential sales from US Energy Information Administration Form 826. We multiply the residential sales proportion, by the total (that is, combined residential and commercial) hourly Federal Energy Regulatory Commission electricity sales to obtain hourly estimates of residential electricity consumption. We sum then aggregate the estimated hourly sales to daily sales for equal temporal comparison to the temperature data. When we fit a regression curve to predict electricity consumption from daily maximum temperature (Supplementary Fig. 2), we find that a quadratic relationship provides the best correlation coefficient without overfitting the data (Supplementary Fig. 3). Using the second-order model, we then forecast electricity consumption between 2020 and 2060 using the future average temperature data for each RCP as the independent variable (Supplementary Fig. 4).

Next, we make some modifications to our archetype model in order to compare it to the regression model. The regression model has the assumptions of constant building stock and population, so we remove the dynamic elements of the model (population, appliance mix, building turnover and so on) for the purposes of validation. The only change in the archetype model for validation is the weather data. In addition, the archetype model is only run for the CBGs within the LADWP service area in order to be comparable to the validation data. We forecast residential electricity consumption in this constant stock and technology case between 2020 and 2060. We then project the regression model with a 95% confidence interval along with the archetype forecasts with GCM temperature averages for each RCP (Supplementary Fig. 4).

**Data availability.** The data that support the findings of this study are available from the corresponding author upon reasonable request.

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

## Acknowledgements

This material is based upon work supported by several National Science Foundation grants (GRFP-DGE 1311230, SRN 1444755, WSC 1360509 and IMEE 1335556 and 1335640). This research was supported in part by an award from the Department of Energy (DOE) Office of Energy Efficiency and Renewable Energy Science and Technology Policy Fellowships administered by the Oak Ridge Institute for Science and Education (ORISE) for the DOE. ORISE is managed by ORAU under DOE contract number DE-SC0014664. Any opinion, findings and conclusions or recommendations expressed in this material are those of the authors and do not necessarily reflect the views of the National Science Foundation. All opinions expressed in this paper are the author's and do not necessarily reflect the policies and views of DOE, ORAU or ORISE.

## Author contributions

J.L.R. and M.V.C designed the study jointly. J.L.R. performed the analyses and collaborated with M.V.C. in interpreting the results and writing the manuscript.

## Additional information

**Competing interests:** The authors declare no competing financial interests.

