## [Peer Review File · Nature Communications]

Reviewers' Comments:

Reviewer #1 (Remarks to the Author)

Thank you for the opportunity to review this paper. This reviewer believes this paper merits publication, but some minor corrections must be implemented and some additional background information should be offered to provide a proper context for the analysis.

Main comments:

1. The assertion that AB32 mandates reducing greenhouse gas (GHG) emissions by 80% from 1990 levels by 2050 is incorrect. AB32 requires California to bring its GHG emissions to 1990 levels by 2020. A Bill requiring reductions of 40% by 2030 failed to pass the Legislature in 2015 and, currently, there are no legal requirements to reduce emissions after 2020. However, there is an Executive Order that sets a goal of reducing GHG emissions by 80%, but the Executive Order does not create a legal mandate. At the same time, SB350 requires the electricity generating sector to drastically reduce GHG emissions by 2030 (De Leon, 2015) and may indirectly set a goal to reduce GHG emissions in California by 40% by 2030.

2. The authors should disaggregate the reported changes in demand by the factors that determine these changes (e.g., population growth, increased temperature, electrification, and technology penetration). This will isolate the effect of, for example, climate change on the reported increases in cooling energy demand.

3. The authors ignore the energy efficiency programs and standards that California has been implementing since the 1970s. In part, they are the reason that energy demand per capita in the residential and commercial sectors has not increased since the 1980s. These programs will continue in the foreseeable future as described in the Scoping Plan adopted by the Air Resources Board. The California Energy Commission and the California Public Utilities Commission are in charge of implementing aggressive energy efficiency programs in California to comply with AB32 and to achieve the 80% emissions reduction goal. The State has also developed and has started to implement a plan to substantially increase energy efficiency in existing buildings (CEC 2015a).

4. The Energy Commission already considers climate change in its official energy forecasts (CEC, 2015b) and has plans to develop its energy system in a way that is less vulnerable to climate disruptions (CNRA, 2016). For this reason, it is unlikely that the dire outcomes described in the paper would materialize in the future.

Minor comments that do not necessarily require changes in the paper:

5. The use of the BCCA downscaling technique may produce some errors in the estimation of peak demand. It has been shown that BCCA underestimates maximum daily temperatures in the upper end of the distribution because it uses the average climate analogue of about 30 historical conditions (Pierce et al., 2014).

6. The RCPs start the projections after 2005 and there are, therefore, historical data for the last 10 years indicating that global emissions in the recent past follow more closely the RCP8.5 scenario. In addition, more than 150 nations in the world have pledged to reduce emissions by 2030 following their own paths. It would be important to briefly discuss the implications of these developments to energy scenarios described in the paper, but this is not essential.

7. Is the rebound effect a function of electricity rates or electricity bills? If it is the later, the rebound effect may be reduced due to the fact that electrification and increased temperatures may

increase electricity bills.

8. The extreme increase in peak demand may be ameliorated by the fact that PVs are penetrating more and more the residential sector and they can at least partially satisfy cooling demand. In addition, it is not inconceivable to assume that in the future, homes and buildings in California would have PVs with battery storage further reducing peak demand concerns.

9. The calibration of the energy demand models to agree with demand reported in the historical period may introduce spurious effects due the difficulties to simulate actual behavior with engineering models. It would have been ideal to combine engineering models with statistical approaches using actual consumption data of homes and building.

Again, with minor modifications, this reviewer recommends publishing this study in Nature Communications.

References:

CEC (2015a). California's Existing Buildings Energy Efficiency Action Plan.
http://docketpublic.energy.ca.gov/PublicDocuments/15-IEPR-05/TN206015_20150904T153548_Existing_Buildings_Energy_Efficiency_Action_Plan.pdf

CEC (2015b). 2015 Integrated Energy Policy Report.

CNRA (2016). Energy Sector Plan. Safeguarding California: Implementation Plans.
<http://resources.ca.gov/docs/climate/safeguarding/Energy%20Sector%20Plan.pdf>

De Leon. Clean Energy and Pollution Reduction Act of 2015. AB 350.
http://leginfo.legislature.ca.gov/faces/billNavClient.xhtml?bill_id=201520160SB350

Pierce, D. W., D. R. Cayan, and B. L. Thrasher, 2014: Statistical downscaling using localized constructed analogs (LOCA). *J. Hydrometeorology*, v. 15, p. 2558, doi:10.1175/JFM-D-14-0082.1

Reviewer #2 (Remarks to the Author)

The paper reports the development of spatially and temporally resolute energy models to project the future energy consumption in residential building stock in Angeles County, California (LAC) under multiple climate change projections. The building models have been calibrated with prototype residential buildings to predict the accurate energy consumption behaviour in the residential building stock. Development of such scenario based studies can be a good approach to propose and implement the energy policies. However, judging from the content and novelty of the manuscript, there are some issues that, in my opinion, weaken the work.

1. The concept of analysis of building stock model for energy prediction under the climate change is not new, which has been reported by various authors, as listed in the reference 5-6 and 10-15. The only difference here is the change of city.

2. From the calibration of the building models, it seems that the authors calibrated the variation in the total annual electricity use against the actual energy use. However, the calibration should be carried out based on the RMS error. This is because the small error in the large value of annual energy use will still be significant and this can cause large variation in the daily energy consumption behaviour, which is an indication of poor calibration. Furthermore, for the validation purposes, authors considered the electricity cost to calculate the energy consumption during the period of 2006-2010. In such cases, inflation rates in the electricity prices should also be considered.

3. The future energy projection was analysed based on several variables including the energy efficient appliances, population growth, occupancy behaviour etc. however, the study lacks of the implementation of renewable energy systems (such as solar panels and solar water heating etc) which will significantly change the energy consumption behaviour in residential buildings

4. Authors tried to estimate the cost of electricity use under the present and future climate scenarios. However, the electricity charge criteria (flat rate, peak and off-peak rate) has neither be considered nor discussed. It is expected that climate change will have significant effect on the peak electricity consumption behaviour and the consideration of peak and off-peak electricity charging rate will have significant impact on the cost savings.

Reviewer #3 (Remarks to the Author)

The study is interesting and relevant. However, the presentation could use some improvements to be more clear and communicate the results better. In particular, the general tone of the paper seems not to properly reflect the results presented. A better comparison with existing literature is also needed. Moreover, the economic analysis proposed seems to be not necessary and not to add significant value.

Detailed issues that should be addressed are reported below:

- (Major issue) The tone in the article seems to suggest that climate change is responsible of the entire electricity demand increase between 2020 and 206, when really the majority of the demand increase is not driven by climate change, but rather by population/economic growth/other. From Line ~100: Scenario 2 RCP 8.5: +68%. Scenario 2 RCP 2.6: +55%. So "climate change" (really different temperatures in 206 following RCP pathways, the difference between RCP 8.5 and 2.6 is not climate change, strictly speaking) in this scenario is responsible for +13%, while other factors are responsible for +55%. Scenario 1 is + 31 and + 40, so climate change is only responsible for +9%. This kind of decomposition should be made clear for all 4 scenarios in the paper and the tone of the paper should reflect this result, especially in the introduction and results sections. Overall, the results presented seem to suggest that electricity demand increase is a concern for LAC, and aggressive climate policies (e.g. RCP 2.6) could partially mitigate this issue.
- Again on this point, what is the % of electricity consumption due to heating/cooling in LAC area? How much does this change in the 4 scenarios and under the different RCPs?
 - "saturation of cooling technology" is a technical jargon that should be explained to the reader of a general journal such as Nature Communications
 - Line 17 "potentially avoiding the installation of new generation capacity". You later mention the issue of peak demand, which really drives the installation of new capacity. Please make the concept clear here as well, or drop.
- (Major issue) Line 40: model of residential energy use in LAC. While the introduction of the supplementary material section is well written and generally correct, it completely neglects a vast body of literature on bottom-up residential demand modeling that emerged after 2010 both in Europe and the US, including several models that separate HVAC loads and appliances. This literature should be mentioned, and in particular the differences between the proposed model and previous works must be explained.
- Line 77 "which previous studies claim is necessary to meet AB 32 GHG goals". References for previous studies are needed.
- Line 81: Do you mean Section 5?
- Scenario 3-4: what are the assumptions for improving building shell? This is a major component of building energy efficiency that seems to have been completely neglected (potentially major issue in the study. If no improvement is assumed this should at least be clearly stated, and I would strongly suggest to re-run including this energy efficiency component and clearly explain improvement assumptions across scenarios).
- (Major issue) model validation: the calibration procedure is pretty clear. However, the validation section could use some work, as it is not completely clear as currently written.

-Line 122: some reference is needed for grid operation and reliability. Moreover, for residential demand, HVAC load is critical as the peak HVAC consumption (hot summer hours) coincide with the overall electricity peak. In scenarios involving electrification, energy efficiency, and aggressive climate policies (such as RCP 2.6 or 4.5), other measures are also likely to occur, such as behaviors change, different HVAC operation strategies (pre-cooling), and generally use of residential demand response mechanisms. There exists a vast literature on all such topics, that should be mentioned in the paper (especially review articles). In particular, use of residential demand response might drastically change the peak demand results shown in the paper, which drives capacity expansion. This caveat must be mentioned.

-Line 143: "Areas away from the coast experience the highest increases in temperatures and subsequent energy use under higher RCPs, but under low RCPs, these are the areas where net savings could be realized". Is this because the majority of the electricity demand increase in these areas is from increased HVAC? Is there a spatial correlation on how HVAC share of total electricity demand evolves over time (this can be reported in a table/figure in the supplemental material)? If so has this been seen in historical years or is this a model artifact/result?

-Line 147 "building shell improvement": can you detail on this? It is listed as an efficiency upgrade here, but it is not clear how this is handled in your modeling.

- Line 148 "maximize energy reduction per dollar invested in efficiency programs". I agree with this statement, but this has not been clearly shown in the paper.

- (major comment) The economic analysis appears in the paper almost out of nowhere. Some more details must be provided to the reader, if this part were to be presented. Also you assume constant cost of electricity across time and scenarios (both 1,2,3,4 and RCPs), which is an assumption that does not hold (in RCP 2.6 by 2060 models - see IPCC AR5 - predicts a carbon price of several hundreds of dollars, which clearly has a significant effect on the price of electricity). This issue should be resolved. I would suggest to drop the economic analysis, which doesn't seem to add much value (the paper is already fairly long and rich in information and details, this part seems to be not necessary and misleading)

-It would be useful to have a table summarizing the assumptions in the 4 scenarios considered, including numerical details (how much more efficient? Building shell assumptions, turnover assumptions, etc.)

-S2: what are the RCPs? What is the purpose of such pathways and who proposed them? (see <http://tntcat.iiasa.ac.at/RcpDb/dsd?Action=htmlpage&page=welcome> for useful information and official list of references)

RESPONSES TO REVIEW 1

Thank you for the opportunity to review this paper. This reviewer believes this paper merits publication, but some minor corrections must be implemented and some additional background information should be offered to provide a proper context for the analysis.

Main comments:

1. The assertion that AB32 mandates reducing greenhouse gas (GHG) emissions by 80% from 1990 levels by 2050 is incorrect. AB32 requires California to bring its GHG emissions to 1990 levels by 2020. A Bill requiring reductions of 40% by 2030 failed to pass the Legislature in 2015 and, currently, there are no legal requirements to reduce emissions after 2020. However, there is an Executive Order that sets a goal of reducing GHG emissions by 80%, but the Executive Order does not create a legal mandate. At the same time, SB350 requires the electricity generating sector to drastically reduce GHG emissions by 2030 (De Leon, 2015) and may indirectly set a goal to reduce GHG emissions in California by 40% by 2030.

Thank you for pointing out this important distinction. We have updated the text to accurately reflect the current policies.

2. The authors should disaggregate the reported changes in demand by the factors that determine these changes (e.g., population growth, increased temperature, electrification, and technology penetration). This will isolate the effect of, for example, climate change on the reported increases in cooling energy demand.

This is an important point raised by other reviewers as well. We now have a table of disaggregated results in the text.

3. The authors ignore the energy efficiency programs and standards that California has been implementing since the 1970s. In part, they are the reason that energy demand per capita in the residential and commercial sectors has not increased since the 1980s. These programs will continue in the foreseeable future as described in the Scoping Plan adopted by the Air Resources Board. The California Energy Commission and the California Public Utilities Commission are in charge of implementing aggressive energy efficiency programs in California to comply with AB32 and to achieve the 80% emissions reduction goal. The State has also developed and has started to implement a plan to substantially increase energy efficiency in existing buildings (CEC 2015a).

This is an important point that this was not clearly discussed in the text. We agree that the state of California has been a leader in implementing energy efficiency, and that this has had a major impact on consumption both within and outside of the state. To contextualize our study in the ongoing efforts in California, we now include a discussion on previous and ongoing efficiency efforts by the state in the introduction.

4. The Energy Commission already considers climate change in its official energy forecasts (CEC, 2015b) and has plans to develop its energy system in a way that is less vulnerable to climate disruptions (CNRA, 2016). For this reason, it is unlikely that the dire outcomes described in the paper would materialize in the future.

The California Energy Demand (CED) forecast model varies from our work in several ways. First and most importantly, it is a short-term model, only capturing changes in demand through 2026, whereas we forecast through 2060. Through the additional 35 years considered in our

study, there is the potential for substantial change in residential energy demand that would not be captured in a 10 year model. Second, the CED is a top-down model, which is appropriate for short-term forecasting, but it would not necessarily hold with major changes in the base conditions (e.g. large population increase or technology turnover). Third, while the CED includes changes from climate change, it does so based on cooling degree days and heating degree days in an econometric model, which is much more simplistic than the RCP models, and might not accurately reflect future trends since it is based on *historic* consumption by HDD and CDD, not bottom-up drivers from the infrastructure itself (as our model is). With regards to California's proposed improvements to the electricity supply (in the energy sector plan), this underscores the fact that climate change could have a major impact on energy use. We think that our work only reinforces the need for long-term supply resilience planning and validates the effort taken by the state of California. Furthermore, the section of the plan that deals with electricity demand recommends incentivizing energy efficiency which is the foundation of the proposed measures from our study. For these reasons, we view our work as complementary to the other research and modeling efforts undertaken by the CEC as it provides longer-term estimates than are currently being considered.

Minor comments that do not necessarily require changes in the paper:

5. The use of the BCCA downscaling technique may produce some errors in the estimation of peak demand. It has been shown that BCCA underestimates maximum daily temperatures in the upper end of the distribution because it uses the average climate analogue of about 30 historical conditions (Pierce et al., 2014).

Thank you for the additional background and citation on this. We have updated the text to reflect that our results might be underestimating annual maximum peak demand.

6. The RCPs start the projections after 2005 and there are, therefore, historical data for the last 10 years indicating that global emissions in the recent past follow more closely the RCP8.5 scenario. In addition, more than 150 nations in the world have pledged to reduce emissions by 2030 following their own paths. It would be important to briefly discuss the implications of these developments to energy scenarios described in the paper, but this is not essential.

Thank you for this suggestion. We've added some discussion on these points.

7. Is the rebound effect a function of electricity rates or electricity bills? If it is the later, the rebound effect may be reduced due to the fact that electrification and increased temperatures may increase electricity bills.

In our model, we assume that electricity rates remain constant with inflation, so we do not attempt to alter the rebound in response to economic factors. Instead, we use a constant 10% energy efficiency penalty (per dx.doi.org/10.1093/reep/rev017). You are correct in saying that electricity will likely be more expensive in the future and will impact efficiency rebound, but forecasting electricity bill changes was beyond the scope of this study. We've added in a sentence mentioning this point.

8. The extreme increase in peak demand may be ameliorated by the fact that PVs are penetrating more and more the residential sector and they can at least partially satisfy cooling demand. In addition, it is not inconceivable to assume that in the future, homes and buildings in California would have PVs with battery storage further reducing peak demand concerns.

PV and onsite storage are increasingly important factors that we haven't captured in our model, and they certainly will have an impact on peak demand. We've included a section covering some of these implications in the discussion.

9. The calibration of the energy demand models to agree with demand reported in the historical period may introduce spurious effects due to the difficulties to simulate actual behavior with engineering models. It would have been ideal to combine engineering models with statistical approaches using actual consumption data of homes and buildings.

We agree that a hybrid approach could have been a useful approach for this study. Lacking the data for this, however, we still think that our conclusions are valid with the engineering model we developed.

Again, with minor modifications, this reviewer recommends publishing this study in *Nature Communications*.

Thank you again for your helpful comments. They have greatly strengthened the manuscript.

RESPONSES TO REVIEW 2

The paper reports the development of spatially and temporally resolute energy models to project the future energy consumption in residential building stock in Angeles County, California (LAC) under multiple climate change projections. The building models have been calibrated with prototype residential buildings to predict the accurate energy consumption behaviour in the residential building stock. Development of such scenario based studies can be a good approach to propose and implement the energy policies. However, judging from the content and novelty of the manuscript, there are some issues that, in my opinion, weaken the work.

Thank you for your review of our work and your proposed improvements.

1. The concept of analysis of building stock model for energy prediction under the climate change is not new, which has been reported by various authors, as listed in the reference 5-6 and 10-15. The only difference here is the change of city.

We agree that energy forecasting is not new, but we do think that our model is an improvement upon previous efforts. For example, there are bottom-up engineering forecast models, but the majority of these include only a small number of building archetypes, and the archetypes themselves generally are simple, single-zone buildings. In our study, we develop a rich ensemble of archetypes to capture the heterogeneity of the residential building stock. Additionally, we couple our ensemble with 10 different GCMs from each of the RCPs to capture the uncertainty from climate forecasts. We have not seen another building forecasting study that has included this many GCMs. We have updated the introduction to clarify the specific contributions of our modeling efforts.

2. From the calibration of the building models, it seems that the authors calibrated the variation in the total annual electricity use against the actual energy use. However, the calibration should be carried out based on the RMS error. This is because the small error in the large value of annual energy use will still be significant and this can cause large variation in the daily energy consumption behaviour, which is an indication of poor calibration. Furthermore, for the validation purposes, authors considered the

electricity cost to calculate the energy consumption during the period of 2006-2010. In such cases, inflation rates in the electricity prices should also be considered.

Thank you for your suggestion on utilizing RMS error to increase sensitivity to temporal variations/outliers in energy use. Unfortunately, we don't have daily electricity data for calibrating the model. We do, however, have electricity data which is spatially differentiated by census block group. We have re-done the calibration of the model utilizing RMS error between the census block groups. This is still annually, but it does provide increased sensitivity to differences throughout different parts of the county.

For the validation, we controlled for electricity price inflation per your suggestion. However, this factor was found to be insignificant when included in the validation model (low t-value and minimal change in regression coefficients). We therefore have not included it in the final validation. The updated validation can be found in the Supporting Information.

3. The future energy projection was analysed based on several variables including the energy efficient appliances, population growth, occupancy behaviour etc. however, the study lacks of the implementation of renewable energy systems (such as solar panels and solar water heating etc) which will significantly change the energy consumption behaviour in residential buildings

This important point was also raised by another reviewer. We now include a large discussion section on the influence of PV, solar water heating, and on-site energy storage in behavior and larger energy trends in the context of our modeling.

4. Authors tried to estimate the cost of electricity use under the present and future climate scenarios. However, the electricity charge criteria (flat rate, peak and off-peak rate) has neither be considered nor discussed. It is expected that climate change will have significant effect on the peak electricity consumption behaviour and the consideration of peak and off-peak electricity charging rate will have significant impact on the cost savings.

We agree that electricity pricing has a large impact on behavior, and peak / off-peak rates will influence both consumer cost and electricity savings. We have decided to remove the cost analysis from the main manuscript as it detracts from the major points of our work and will perhaps expand these ideas further in a subsequent article.

RESPONSES TO REVIEW 3

The study is interesting and relevant. However, the presentation could use some improvements to be more clear and communicate the results better. In particular, the general tone of the paper seems not to properly reflect the results presented. A better comparison with existing literature is also needed. Moreover, the economic analysis proposed seems to be not necessary and not to add significant value.

Detailed issues that should be addressed are reported below:

(Major issue) The tone in the article seems to suggest that climate change is responsible of the entire electricity demand increase between 2020 and 206, when really the majority of the demand increase is not driven by climate change, but rather by population/economic growth/other. From Line ~100: Scenario 2 RCP 8.5: +68%. Scenario 2 RCP 2.6: +55%. So "climate change" (really different temperatures in 206 following RCP pathways, the difference between RCP 8.5 and 2.6 is not climate change, strictly speaking) in this scenario is responsible for +13%, while other factors are responsible for +55%. Scenario 1 is + 31 and + 40, so climate change is only responsible for +9%. This kind of

decomposition should be made clear for all 4 scenarios in the paper and the tone of the paper should reflect this result, especially in the introduction and results sections. Overall, the results presented seem to suggest that electricity demand increase is a concern for LAC, and aggressive climate policies (e.g. RCP 2.6) could partially mitigate this issue.

Thank you for this suggestion. We now include a table disaggregating results by major drivers (population, climate change, etc.). We have also updated the tone to accurately reflect the findings of our study.

- Again on this point, what is the % of electricity consumption due to heating/cooling in LAC area? How much does this change in the 4 scenarios and under the different RCPs?

We have updated the end-use consumption tables, so that this information is more readily available to the reader. We have also updated the results section to include more of this information.

- "saturation of cooling technology" is a technical jargon that should be explained to the reader of a general journal such as Nature Communications

We have changed the language to more common terms in most places and defined this term during the saturation discussion.

- Line 17 "potentially avoiding the installation of new generation capacity". You later mention the issue of peak demand, which really drives the installation of new capacity. Please make the concept clear here as well, or drop.

We have clarified the text to fully explain this concept.

- (Major issue) Line 40: model of residential energy use in LAC. While the introduction of the supplementary material section is well written and generally correct, it completely neglects a vast body of literature on bottom-up residential demand modeling that emerged after 2010 both in Europe and the US, including several models that separate HVAC loads and appliances. This literature should be mentioned, and in particular the differences between the proposed model and previous works must be explained.

Thank you for bringing this body of literature to our attention. We have re-written the literature review and moved it into the main text. In this section we highlight the contribution of our study in the context of previous work.

-Line 77 "which previous studies claim is necessary to meet AB 32 GHG goals". References for previous studies are needed.

The sentence has been updated and the studies cited appropriately.

-Line 81: Do you mean Section 5?

Yes, thank you for catching this. The manuscript and SI have been restructured during the revision, and all cross-references have been updated appropriately.

- Scenario 3-4: what are the assumptions for improving building shell? This is a major component of building energy efficiency that seems to have been completely neglected (potentially major issue in the

study. If no improvement is assumed this should at least be clearly stated, and I would strongly suggest to re-run including this energy efficiency component and clearly explain improvement assumptions across scenarios).

To capture improvements in building shells over time, older building archetypes were replaced with newer, more efficient archetypes. The turnover rates in the base cases were based upon our previous modeling of Los Angeles County (Reyna & Chester 2015, Journal of Industrial Ecology), and these rates were augmented to emulate shell improvements (insulation, windows, etc.) This approach saves greatly on computation time over individual quantification of the shell components at the expense of end-use transparency since newer archetypes have these improvements "bundled" together. We have updated the text to more clearly describe the turnover modeling and assumptions and moved some of the descriptive text from the Supporting Information.

- (Major issue) model validation: the calibration procedure is pretty clear. However, the validation section could use some work, as it is not completely clear as currently written.

We have updated the description of the validation. We've tried to clarify both our procedure and our data sources.

-Line 122: some reference is needed for grid operation and reliability. Moreover, for residential demand, HVAC load is critical as the peak HVAC consumption (hot summer hours) coincide with the overall electricity peak. In scenarios involving electrification, energy efficiency, and aggressive climate policies (such as RCP 2.6 or 4.5), other measures are also likely to occur, such as behaviors change, different HVAC operation strategies (pre-cooling), and generally use of residential demand response mechanisms. There exists a vast literature on all such topics, that should be mentioned in the paper (especially review articles). In particular, use of residential demand response might drastically change the peak demand results shown in the paper, which drives capacity expansion. This caveat must be mentioned.

These are important factors that influence residential energy demand, and thank you for pointing this out. In our revised discussion, we now include a section on other factors that could influence the results of our study, such as behavioral changes and renewable energy technologies.

-Line 143: "Areas away from the coast experience the highest increases in temperatures and subsequent energy use under higher RCPs, but under low RCPs, these are the areas where net savings could be realized". Is this because the majority of the electricity demand increase in these areas is from increased HVAC? Is there a spatial correlation on how HVAC share of total electricity demand evolves over time (this can be reported in a table/figure in the supplemental material)? If so has this been seen in historical years or is this a model artifact/result?

Yes, these differences are driven by an increased share of HVAC. Overall, HVAC accounts for 60-80% of consumption increases, depending on the scenario, RCP, and location. We've included spatial results in the Supporting Information. We only have one year of electricity data with spatial differentiation for comparison so we cannot confirm for sure if the spatial differentiation of HVAC drivers is a result of the way our model is set up, but due to the large differences in climate across Los Angeles County, we would speculate that this is not a model artifact.

-Line 147 "building shell improvement": can you detail on this? It is listed as an efficiency upgrade here, but it is not clear how this is handled in your modeling.

In our archetypes, newer vintage buildings have higher efficiency shells (e.g. more insulation, better quality windows, etc.). To capture building shell improvements over time, we have newly constructed buildings or turned over buildings replaced with the newest (i.e. most efficient-shelled) building within each category. For example, 1970s single family buildings in climate zone 9 would be replaced with climate zone 9 post-2000 vintage buildings. In the model, this does peg the upper limit of building shell efficiency to what is currently in the stock. We have expanded the text on this subject to cover the turnover modeling process and how this stands in as an approximation for building shell improvements.

- Line 148 "maximize energy reduction per dollar invested in efficiency programs". I agree with this statement, but this has not been clearly shown in the paper.

The language in this sentence has been updated to not over-reach the scope of the paper.

- (major comment) The economic analysis appears in the paper almost out of nowhere. Some more details must be provided to the reader, if this part were to be presented. Also you assume constant cost of electricity across time and scenarios (both 1,2,3,4 and RCPs), which is an assumption that does not hold (in RCP 2.6 by 2060 models - see IPCC AR5 - predicts a carbon price of several hundreds of dollars, which clearly has a significant effect on the price of electricity). This issue should be resolved. I would suggest to drop the economic analysis, which doesn't seem to add much value (the paper is already fairly long and rich in information and details, this part seems to be not necessary and misleading)

We agree that the economic analysis was distracting from the major conclusions of the paper, so we have removed it from the main text as it is better suited as a more in-depth stand alone study.

-It would be useful to have a table summarizing the assumptions in the 4 scenarios considered, including numerical details (how much more efficient? Building shell assumptions, turnover assumptions, etc.)

We have moved our scenario summary table from the SI to the main text and added in a few more details on the scenarios.

-S2: what are the RCPs? What is the purpose of such pathways and who proposed them? (see <http://tntcat.iiasa.ac.at/RcpDb/dsd?Action=htmlpage&page=welcome> for useful information and official list of references)

We have now included some of your suggested background information on the RCPs into our methodology.

Reviewers' Comments:

Reviewer #1 (Remarks to the Author)

Thank for adequately addressing my comments.

The Governor recently signed into law SB32, which mandates reducing GHG emissions by 40% by 2030. This is only for your information.

https://leginfo.legislature.ca.gov/faces/billNavClient.xhtml?bill_id=201520160SB32

In my opinion, this paper will be an excellent contribution to the scientific literature on energy demand and climate change.

Reviewer #2 (Remarks to the Author)

I believe the authors have addressed all concerns raised in our review. It can be considered for publication.